# Comparison of Amoxicillin Administered Twice versus Four Times a Day in First-Line *Helicobacter pylori* Eradication Using Tegoprazan, Clarithromycin, and Bismuth: A Propensity Score Matching Analysis

**DOI:** 10.3390/microorganisms12101952

**Published:** 2024-09-27

**Authors:** Jun-Hyung Cho, So-Young Jin

**Affiliations:** 1Digestive Disease Center, Soonchunhyang University Hospital, Yongsan-gu, Seoul 04401, Republic of Korea; 2Department of Pathology, Soonchunhyang University Hospital, Yongsan-gu, Seoul 04401, Republic of Korea; jin0924@schmc.ac.kr

**Keywords:** *Helicobacter pylori*, eradication, amoxicillin, dosing schedule, bismuth

## Abstract

This study aimed to investigate the effects of different amoxicillin (AMX) dosing schedules on bismuth quadruple therapy in *Helicobacter pylori* treatment-naïve patients. A total of 139 *H. pylori*-infected patients received a 2-week eradication regimen consisting of 50 mg tegoprazan, 500 mg clarithromycin, and 300 mg bismuth tripotassium dicitrate twice daily, 1000 mg AMX twice daily (BID group), or 500 mg AMX four times daily (QID group). We performed a urea breath test to evaluate *H. pylori* eradication eight weeks after treatment and compared the *H. pylori* eradication rate, patient compliance, and adverse drug events between the BID and QID groups. Based on propensity score matching, 114 and 100 patients were included in intention-to-treat (ITT) and per-protocol (PP) analyses, respectively. The *H. pylori* eradication rate did not differ significantly according to the ITT (82.5% vs. 87.7%, *p* = 0.429) and PP (95.9% vs. 98.0%, *p* = 0.536) analyses between the BID and QID groups. No significant differences were found in treatment compliance or adverse drug event rates between the two groups. In conclusion, the eradication rate of first-line *H. pylori* therapy containing tegoprazan, clarithromycin, and bismuth was not affected by AMX dosing schedules administered twice and four times daily.

## 1. Introduction

*Helicobacter pylori* infection is a major cause of peptic ulcers and gastric cancer [1]. *H. pylori* therapy can reduce the recurrence of peptic ulcers and prevent gastric cancer development. Traditionally, standard triple therapy consisting of a standard dose of a proton pump inhibitor (PPI), 1000 mg of amoxicillin (AMX), and 500 mg of clarithromycin twice daily has been used to eradicate *H. pylori* [2]. Standard triple therapy for 1–2 weeks has long been effective as a first-line *H. pylori* eradication method. However, eradication rates have decreased due to increased antibiotic resistance to clarithromycin [3]. Among the antimicrobial agents used for *H. pylori* treatment, the AMX resistance rate is relatively low at 5–10% in Asia, Europe, and America [4].

AMX, a beta-lactamase antibiotic, binds to and inactivates penicillin-binding proteins, weakening the bacterial cell wall and causing cell lysis [5]. It has a short plasma half-life at approximately 1–2 h and a time-dependent bactericidal effect. In microbiology, the minimum inhibitory concentration (MIC) is the lowest antibiotic concentration that prevents bacterial growth in vitro [6]. Frequent AMX dosing has the clinical advantage of maintaining plasma concentrations higher than the MIC; theoretically, AMX 500 mg administered four times daily can maximize the time above the MIC for up to 24 h [7]. Nevertheless, *H. pylori*-infected patients received AMX 1000 mg twice daily as standard triple therapy.

Potassium-competitive acid blockers (P-CABs) and bismuth may serve as alternatives to other antibiotics to overcome the unsatisfactory results of conventional triple therapy [8,9]. In a previous study, we evaluated the efficacy of a 2-week quadruple therapy (tegoprazan 50 mg + AMX 1000 mg + clarithromycin 500 mg + bismuth tripotassium dicitrate 300 mg twice daily) in *H. pylori* treatment-naïve patients [10]. The eradication success rates were 82.9% and 95.8% in intention-to-treat (ITT) and per-protocol (PP) analyses, respectively. This study design compared *H. pylori* eradication, patient compliance, and adverse drug events between AMX 1000 mg twice daily and 500 mg four times daily in tegoprazan-based bismuth quadruple therapy to determine whether the frequent dosing schedule of AMX affects *H. pylori* treatment.

## 2. Materials and Methods

### 2.1. Patients and Study Design

We retrospectively reviewed the medical records of *H. pylori*-infected patients treated with a 2-week tegoprazan-based quadruple regimen containing AMX, clarithromycin, and bismuth at a single center from March 2022 to May 2024. From March 2023, the AMX dosing schedule was changed from 1000 mg twice daily to 500 mg four times daily to increase the *H. pylori* eradication rate. The reasons for *H. pylori* eradication include peptic ulcers, endoscopic resection of gastric neoplasia, and chronic active gastritis. The exclusion criteria were an age < 20 or >80 years, history of gastric surgery, previous *H. pylori* eradication, severe systemic disease, or other known causes. The study protocol was approved by the Institutional Review Board of our hospital (SCHUH 2024-04-011).

### 2.2. Helicobacter pylori Eradication Therapy

All patients received a 2-week first-line *H. pylori* eradication regimen consisting of 50 mg tegoprazan, 500 mg clarithromycin, and 300 mg bismuth tripotassium dicitrate twice daily, 1000 mg AMX twice daily (BID group), or 500 mg AMX four times daily (QID group). Six weeks after completing eradication therapy, the patients were followed up to assess compliance and adverse drug events. The eradication result was confirmed using the urea breath test (UBT) with a cut-off value of 2.5‰, as recommended by the manufacturer. All patients were instructed to discontinue PPIs and P-CABs for at least 2 weeks, as well as antibiotics and bismuth for at least 1 month before undergoing UBT.

### 2.3. Study Outcomes

The primary outcome of this study was *H. pylori* eradication rates in the BID and QID groups. In the ITT analysis, patients who were noncompliant or lost to follow-up were considered to have failed the *H. pylori* treatment. We performed the PP analysis after excluding patients who did not complete the eradication regimen or did not follow up. Secondary endpoints were patient compliance and adverse drug events, including bitter taste, abdominal pain, nausea or vomiting, diarrhea, and bloating. A physician (J-H Cho) evaluated treatment compliance and adverse drug events when the patients returned to the outpatient clinic to assess eradication success. Compliance with the eradication regimen was defined as more than 90% consumption of the medication.

### 2.4. Statistical Analysis

The eradication rate of 2-week quadruple therapy was 83% based on our previous study [10]. We assumed a difference of 12% between eradication rates achieved by the two regimens. The minimum sample size was calculated using α- and β-error values of 0.05 and 0.2, respectively. The calculated sample size was 94 in each group, considering a drop-out rate of 10%.

Continuous variables are presented as mean ± standard deviation and were compared using Student’s *t*-test. Categorical variables are presented as numbers with percentages and were compared using Pearson’s chi-square test. The retrospective nature of this study incorporated potential confounding and treatment-related selection biases between the two groups. To balance the two treatment groups, propensity score matching at a ratio of 1:1 was performed using a logistic regression model for covariates such as age, sex, alcohol intake, smoking status, body mass index (BMI), cause of need for eradication, and comorbidities. All statistical analyses were conducted using SPSS software (version 27.0; IBM Corp., Armonk, NY, USA). A *p*-value < 0.05 was considered significant.

## 3. Results

### 3.1. Study Population

Figure 1 presents a flowchart of patient enrollment in this study. After excluding three patients with a history of gastric surgery (*n* = 1) and those who refused follow-up UBT (*n* = 2), 136 patients were assessed for eligibility. In terms of AMX dosage and frequency of administration, 63 and 73 patients received 1000 mg AMX twice daily (BID group) or 500 mg AMX four times daily (QID group), respectively. We included 114 patients (57 in each group) in the ITT analysis through propensity score matching. After excluding 2 noncompliant patients and 12 who did not follow up, 49 and 51 patients in the BID and QID groups, respectively, remained for the PP analysis.

The baseline patient characteristics are presented in Table 1. Before propensity score matching, a marginal difference was observed between the BID and QID groups regarding current smoking habits (*p* = 0.055). However, all baseline characteristics after propensity score matching were similar between the groups. The mean age was 53.4 ± 12.6 and 54.8 ± 11.9 years in the BID and QID groups, respectively (*p* = 0.538). The proportion of male patients was 54.4% in both groups (*p* = 1.000). No significant differences in alcohol intake, smoking status, BMI, cause of need for eradication, or comorbidities were observed between the groups.

### 3.2. Eradication Rate

Table 2 shows the first-line *H. pylori* eradication rates of the two groups. In the ITT analysis, the eradication rates of *H. pylori* were 82.5% (*n* = 47/57) and 87.7% (*n* = 50/57) in the BID and QID groups, respectively. In the PP analysis, the eradication rates were 95.9% (*n* = 47/49) in the BID group and 98.0% (*n* = 50/51) in the QID group. The eradication rates did not differ between the two groups in the ITT (*p* = 0.429) and PP analyses (*p* = 0.536).

### 3.3. Compliance and Adverse Events

Patient compliance and adverse drug events are summarized in Table 3. In total, 98.0% of patients in the BID group and 98.1% in the QID group adhered to the eradication regimen. Treatment compliance between the groups did not differ significantly (*p* = 1.000). Adverse events associated with *H. pylori* eradication therapy were observed in 20 (40.0%) and 12 (23.1%) patients in the BID and QID groups, respectively (*p* = 0.066). The perception of bitter taste was the most common, accounting for 26.0% of the BID group and 17.3% of the QID group (*p* = 0.286). The incidence of nausea and/or vomiting was higher in the BID group (14.0%) than in the QID group (3.8%); however, this difference was marginally insignificant (*p* = 0.089). No significant differences were found in the frequencies of abdominal pain, diarrhea, bloating, or others between the two groups. No serious adverse events were observed in either group.

### 3.4. Analysis of Clinical Factors Affecting H. pylori Eradication

We investigated the various clinical factors that affect *H. pylori* eradication (Table 4). In relation to *H. pylori* eradication rate, no significant difference was found in terms of sex, age, BMI, smoking status, cause of need for eradication, presence of comorbidity, or occurrence of adverse drug events. The *H. pylori* eradication rate in alcohol drinkers was lower than in non-drinkers; however, this difference was not statistically significant (93.0% vs. 100%, *p* = 0.076).

## 4. Discussion

In an era of increasing antibiotic resistance, *H. pylori* eradication rates of conventional treatments have reached unacceptable levels. At 30–40%, the clarithromycin resistance rate of *H. pylori* shows unsatisfactory results with standard triple therapy in some countries [11]. The Maastricht VI consensus report recommends tailored *H. pylori* therapy or empirical regimens (classic bismuth quadruple and concomitant therapies) in areas of clarithromycin-resistant *H. pylori* rates greater than 15% [12]. Tailored therapy requires expertise in antimicrobial susceptibility testing before selecting the *H. pylori* eradication regimen [13]. In particular, culture-based *H. pylori* eradication is time consuming and challenging in clinical settings. Classic quadruple therapies, including tetracycline and metronidazole, are reportedly associated with a high incidence of adverse drug events [14]. Treatment compliance may decrease because of the high pill burden in patients receiving concomitant therapy with the three antibiotics, and if eradication is not achieved, secondary antibiotic resistance may develop in *H. pylori* [15].

In the latest South Korean guidelines for *H. pylori* treatment, empirical *H. pylori* treatment comprises three first-line regimens: standard triple therapy for 2 weeks, concomitant or sequential therapy for 10 days, and bismuth quadruple therapy for 10–14 days [16]. However, the pooled eradication rate of the 2-week standard triple therapy was 78.1%, which did not meet the international standards for first-line *H. pylori* treatment success rates. Classic quadruple and concomitant therapies are not the preferred first-line therapies, resulting in incredibly low prescription rates of 2.6% and 3.1%, respectively [17].

AMX is one of the most frequently used antibiotics for *H. pylori* eradication owing to its low resistance level. Since AMX is a time-dependent antibiotic, its antibacterial effect depends on whether its plasma concentration is maintained above the MIC. The standard administration schedule of AMX for other infectious diseases is three to four times a day [18]. An MIC of 1 μg/mL was reached 83.3% of the time with four-times-daily doses of AMX compared to 45.8% with the same total amount given twice daily [19]. Two previous studies evaluated the therapeutic efficacy of different AMX dosing schedules in PPI-based standard triple therapy. In a South Korean study, Kim et al. compared *H. pylori* eradication between AMX 1000 mg twice daily and AMX 500 mg four times daily for 2 weeks [20]. Regarding eradication rates, AMX 500 mg four times daily was not superior to AMX 1000 mg twice daily (ITT, 89.2% vs. 91.4%, *p* = 0.620; PP, 90.8% vs. 92.1%, *p* = 0.752). In a Japanese study, patients infected with clarithromycin-susceptible *H. pylori* strains were treated with a 1-week standard triple therapy containing three AMX dosing schedules (750 mg twice daily, 500 mg three times daily, and 500 mg four times daily) [21]. When AMX 500 mg was administered three or four times daily, the *H. pylori* eradication rate was higher than when AMX 750 mg was administered twice daily (ITT, 91.9–93.5% vs. 77.8%, *p* < 0.05; PP, 95.0–96.7% vs. 80.3%, *p* < 0.05, respectively). These inconsistent results may have been influenced by patient characteristics and antibiotic resistance rates in different regions. However, these studies were conducted before P-CAB development.

P-CABs have a faster onset of action and a stronger effect on suppressing gastric acid secretion than conventional PPIs [22]. When P-CAB increases the intragastric pH to >6 within four hours, *H. pylori* may be in the replication phase and thus more susceptible to antibiotics [23]. Owing to the acid lability of AMX, P-CAB may improve its stability and bioavailability in the stomach. Bismuth is a semi-metal that inhibits protein and cell wall synthesis in *H. pylori* [24]. In human studies, bismuth addition increased the eradication rate of resistant *H. pylori* strains by 30–40% [25]. A real-world evidence study showed that the bismuth addition improved *H. pylori* eradication compared to no bismuth addition in a 2-week P-CAB-based triple therapy (ITT, 82.9% vs. 71.8%, *p* = 0.029; PP, 95.8% vs. 87.5%, *p* = 0.227) [10].

The dose and administration schedule of antibiotics should be determined based on their pharmacological properties to optimize bacterial infection treatment [26]. Nevertheless, most *H. pylori* treatments containing AMX are prescribed twice daily. To our knowledge, no previous study has evaluated *H. pylori* eradication using tegoprazan, clarithromycin, and bismuth according to AMX dosing schedules. This study compared the treatment efficacy of AMX 1000 mg twice daily (BID group) and AMX 500 mg four times daily (QID group). The overall eradication rates between the two groups were similar (82.5–95.9% in the BID group and 87.7–98.0% in the QID group).

Recently, AMX-based dual therapy has emerged as an *H. pylori* eradication strategy [27]. Potent gastric acid blockers such as P-CAB enable AMX to treat *H. pylori* infection. In AMX-based dual therapy, AMX should be administered at divided doses of 2–3 g/day, preferably three or four times daily rather than twice daily [28]. A 14-day AMX dual therapy is recommended, rather than a 7- or 10-day treatment period [29]. In this study, we hypothesized that a low dose of AMX 2 g/day administered four times might provide an additive effect to dual therapy. However, no benefit was found in the QID group compared with the BID group. Hence, we suggest that AMX 1000 mg twice daily is sufficient when prescribing *H. pylori* eradication therapy consisting of tegoprazan, clarithromycin, and bismuth.

This study had several limitations. First, we did not evaluate the resistance of *H. pylori* to antimicrobial agents. The eradication rates were not assessed based on the presence of clarithromycin susceptibility in *H. pylori*. Second, the intragastric pH and plasma AMX concentration were not measured. Therefore, we could not investigate the correlation between the AMX dosing schedule and the pharmacokinetics of AMX. Third, this was a retrospective study based on data collected at our hospital. Although propensity score matching ensured that the baseline patient characteristics were similar between the groups, a randomized controlled trial (RCT) is needed to confirm our results. Finally, this study included a small number of patients and was conducted at a single center. Thus, the number of patients obtained through sample size calculation was not reached. The eradication outcomes may vary depending on the antibiotic resistance of *H. pylori* in each region of South Korea. Multicenter RCTs with larger sample sizes are required.

## 5. Conclusions

AMX dosing schedules of twice and four times daily did not affect the *H. pylori* eradication outcomes in the 2-week quadruple regimen containing tegoprazan, AMX, clarithromycin, and bismuth for treating *H. pylori*-infected patients.

## Figures and Tables

**Figure 1 microorganisms-12-01952-f001:**
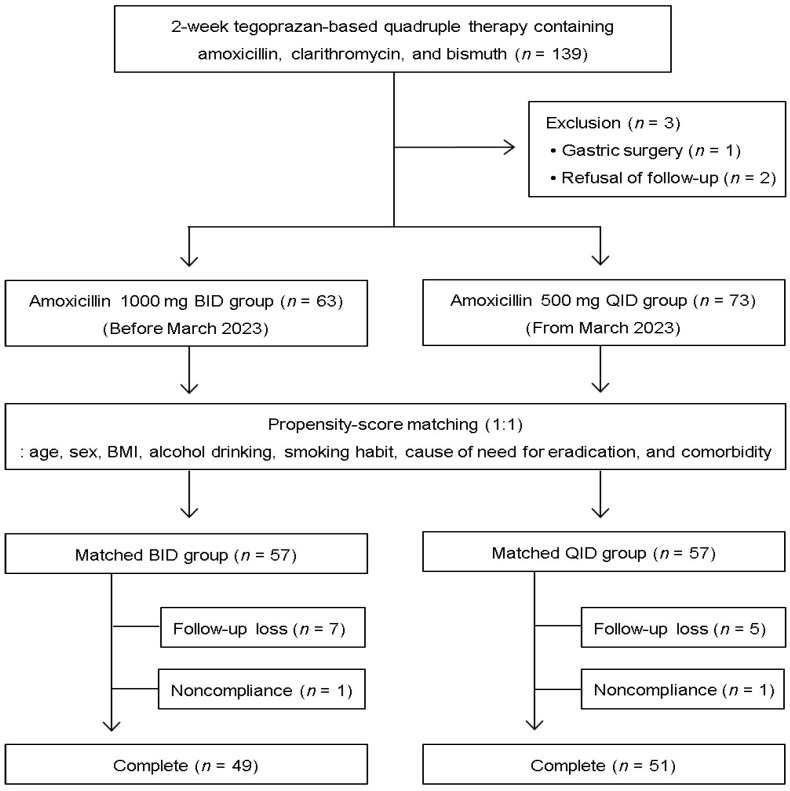
Flowchart for patient enrollment. BMI, body mass index.

**Table 1 microorganisms-12-01952-t001:** Baseline characteristics of patients before and after propensity score matching.

Variable	Before Matching			After Matching		
	BID Group (*n* = 63)	QID Group (*n* = 73)	*p*-Value	BID Group (*n* = 57)	QID Group (*n* = 57)	*p*-Value
Age (years)	53.3 ± 12.2	55.5 ± 11.5	0.286	53.4 ± 12.6	54.8 ± 11.9	0.538
Male (%)	35 (55.6)	35 (47.9)	0.376	31 (54.4)	31 (54.4)	1.000
Alcohol drinking (%)	28 (44.4)	32 (43.8)	0.943	26 (45.6)	23 (40.4)	0.570
Current smoker (%)	18 (28.6)	11 (15.1)	0.055	13 (22.8)	11 (19.3)	0.646
Body mass index (kg/m^2^)	24.2 ± 3.7	24.0 ± 3.3	0.712	24.4 ± 3.7	24.1 ± 3.4	0.611
Cause of need for eradication (%)			0.344			0.404
Chronic active gastritis	41 (65.1)	53 (72.6)		39 (68.4)	43 (75.4)	
Peptic ulcer/neoplasia	22 (34.9)	20 (27.4)		18 (31.6)	14 (24.6)	
Laboratory finding, mean (SD)						
Hemoglobin (g/dL)	13.9 ± 1.8	13.7 ± 1.3	0.529	14.0 ± 1.8	13.9 ± 1.4	0.814
AST (U/L)	23.1 ± 8.7	23.8 ± 7.3	0.600	23.5 ± 9.0	23.7 ± 7.2	0.887
ALT (U/L)	24.4 ± 16.3	22.9 ± 9.8	0.539	24.6 ± 16.9	22.8 ± 9.3	0.495
Creatinine (mg/dL)	0.83 ± 0.17	0.83 ± 0.16	0.944	0.84 ± 0.17	0.85 ± 0.17	0.682
Underlying disease (%)						
Cardiovascular disease	20 (31.7)	24 (32.9)	0.888	18 (31.6)	20 (35.1)	0.691
Respiratory disease	0 (0)	0 (0)	1.000	0 (0)	0 (0)	1.000
Liver dysfunction	0 (0)	2 (2.7)	0.499	0 (0)	2 (3.5)	0.496
Renal dysfunction	1 (1.6)	0 (0)	0.463	1 (1.8)	0 (0)	1.000
Diabetes	7 (11.1)	5 (6.8)	0.382	5 (8.8)	3 (5.3)	0.716

SD, standard deviation; AST, aspartate aminotransferase; ALT, alanine aminotransferase.

**Table 2 microorganisms-12-01952-t002:** Comparison of eradication rates between two- and four-times-daily amoxicillin dosing.

	BID Group	QID Group	Adjusted 95% CI for Difference	*p*-Value
Intention-to-treat analysis				
Eradication rate	82.5% (47/57)	87.7% (50/57)	5.2% (−7.8 to 18.3%)	0.429
Per-protocol analysis				
Eradication rate	95.9% (47/49)	98.0% (50/51)	2.1% (−4.6 to 8.8%)	0.536

CI, confidence interval.

**Table 3 microorganisms-12-01952-t003:** Compliance and adverse events associated with two- and four-times-daily amoxicillin dosing.

	BID Group (*n* = 50)	QID Group (*n* = 52)	*p*-Value
Compliance	49 (98.0%)	51 (98.1%)	1.000
Patients with adverse events	20 (40.0%)	12 (23.1%)	0.066
Bitter taste	13 (26.0%)	9 (17.3%)	0.286
Abdominal pain	1 (2%)	0 (0%)	0.490
Nausea or vomiting	7 (14.0%)	2 (3.8%)	0.089
Diarrhea	3 (6.0%)	0 (0%)	0.114
Bloating	1 (2.0%)	1 (1.9%)	1.000
Others	3 (6.0%)	0 (0%)	0.114

**Table 4 microorganisms-12-01952-t004:** Association between clinical factors and *Helicobacter pylori* eradication.

Variable	Eradication Rate	*p*-Value
Sex		0.245
Male (*n* = 53)	50 (94.3%)	
Female (*n* = 47)	47 (100%)	
Age, years		0.294
<50 (*n* = 36)	34 (94.4%)	
≥50 (*n* = 64)	63 (98.4%)	
Body mass index, kg/m^2^		0.559
<25 (*n* = 61)	60 (98.4%)	
≥25 (*n* = 39)	37 (94.9%)	
Alcohol drinking		0.076
No (*n* = 57)	57 (100%)	
Yes (*n* = 43)	40 (93.0%)	
Current smoker		1.000
No (*n* = 81)	78 (96.3%)	
Yes (*n* = 19)	19 (100%)	
Cause of need for eradication		1.000
Chronic active gastritis (*n* = 73)	71 (97.3%)	
Peptic ulcer/neoplasia (*n* = 27)	26 (96.3%)	
Comorbidity		0.562
Absent (*n* = 60)	59 (98.3%)	
Present (*n* = 40)	38 (95.0%)	
Adverse drug event		0.550
Absent (*n* = 69)	66 (95.7%)	
Present (*n* = 31)	31 (100%)	

## Data Availability

The data that support the findings of this study are available on request from the corresponding author. The data are not publicly available due to privacy or ethical restrictions.

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
