# Peer review of "Comparison of Amoxicillin Administered Twice versus Four Times a Day in First-Line Helicobacter pylori Eradication Using Tegoprazan, Clarithromycin, and Bismuth: A Propensity Score Matching Analysis"

_microorganisms, 2024, doi:10.3390/microorganisms12101952_

Round 1

Reviewer 1 Report

Comments and Suggestions for Authors

In the present retrospective study with propensity score analysis, Cho et al showed that administration of amoxicillin (AMO) 500 mg qid or 1000 mg bid allowed the same eradication rate of bismuth quadruple therapy for H. pylori. Main comments:

1) Sample size calculation is missing. This could be of paramount importance, as the difference between the two groups, although not statistically significant, as about 5%, therefore it is possible that with a larger sample size the results could have been different.

2) As this was a retrospective study, I did not understand how the physician decided to prescribe AMO bid or qid. Indeed, randomization is absent, which is a relevant limitation in this study.

Author Response

  1. Sample size calculation is missing. This could be of paramount importance, as the difference between the two groups, although not statistically significant, as about 5%, therefore it is possible that with a larger sample size the results could have been different.

Answer: Thank you for your valuable comment. As you know, sample size calculations are usually performed in prospective studies to balance baseline patient characteristics between intervention groups. Because the data for this study were collected retrospectively from electronic medical records, we performed a propensity score-matched analysis in a 1:1 ratio. However, we acknowledge the major limitations of this study. Therefore, we mentioned the need for randomized controlled trials to confirm these results in the Discussion section.

  1. As this was a retrospective study, I did not understand how the physician decided to prescribe AMO bid or qid. Indeed, randomization is absent, which is a relevant limitation in this study.

Answer: Thank you for your careful comment. Because this study was not conducted prospectively, the amoxicillin dosing schedule was not randomly assigned to the BID and QID groups. Since March 2023, to increase the eradication rate of H. pylori at our hospital, the amoxicillin administration has been changed from 1000 mg twice daily to 500 mg four times daily. Despite being retrospective data, this is the first study to evaluate H. pylori eradication using tegoprazan, clarithromycin, and bismuth according to amoxicillin dosing schedules. However, this study has limitations in verifying our results. In the future, we plan to conduct a multicenter randomized controlled trial with large sample size.

Reviewer 2 Report

Comments and Suggestions for Authors

Authors of the manuscript entitled “Comparison of Amoxicillin Administered Twice versus Four Times a Day in First-Line Helicobacter pylori Eradication using Tegoprazan, Clarithromycin, and Bismuth: A Propensity Score Matching Analysis” presented a retrospective comparative investigation for microbial effectiveness as well as patient compliance and adverse effects for two AMX regimens within tegoprazan-based bismuth quadruple therapy against H. Pylori. The study is relevant in its field redeeming publication following these comments and suggestions:

1. Authors should highlight the history of antibiotic provision before conducting the study. Was there any relevant antibiotic free-time adopted before conducting the study? Previous or concomitant antimicrobial administration should be highlighted or avoided since this could represent potential covariate.

2. Patient lab signs should be annotated at patients' base line descriptions.

3. The study further lacked more detailed information regarding, patients’ renal/liver status, and lab analysis to assess the pharamcokinetic aspects in terms of efficiency.

4. Authors should conduct multivariate logistic regression to identify the independent factors and covariates (predictors) associated with risk or protective factors for the class microbial outcomes of the employed therapy. Further, plots for conditional estimation/prediction of the multivariate logistic regression analysis should be demonstrated.

5. Finally, in lines 109-110, text should be corrected, as statistical difference should not be highlighted at non-significant difference p <0.1

Author Response

  1. Authors should highlight the history of antibiotic provision before conducting the study. Was there any relevant antibiotic free-time adopted before conducting the study? Previous or concomitant antimicrobial administration should be highlighted or avoided since this could represent potential covariate.

Answer: Thank you for your careful comment. We added the history of antibiotic provision in the Method section as below.

From March 2023, the AMX dosing schedule was changed from 1000 mg twice daily to 500 mg four times daily to increase the H. pylori eradication rate.

In the revised Figure 1, ‘Before March 2023’ and ‘From March 2023’ were added to ‘Amoxicillin 1000 mg BID group’ and ‘Amoxicillin 500 mg QID group’.

All patients were confirmed by a physician (J-H Cho) to have received no other antibiotics prior to H. pylori treatment.

  1. Patient lab signs should be annotated at patients' base line descriptions.

Answer: Thank you for your valuable comment. We have added the patient's laboratory results including serum hemoglobin, AST, ALT, and creatinine levels to Table 1. Therefore, you can check the updated data in the revised manuscript.

  1. The study further lacked more detailed information regarding, patients’ renal/liver status, and lab analysis to assess the pharamcokinetic aspects in terms of efficiency.

Answer: Thank you for your kind comment. In the revised Table 1, we categorized patient`s comorbidities into individual systemic diseases, such as cardiovascular disease, respiratory disease, liver dysfunction, renal dysfunction, and diabetes.  

  1. Authors should conduct multivariate logistic regression to identify the independent factors and covariates (predictors) associated with risk or protective factors for the class microbial outcomes of the employed therapy. Further, plots for conditional estimation/prediction of the multivariate logistic regression analysis should be demonstrated.

Answer: Thank you for your careful comment. In this study, a small number of patients (n = 3/100) failed to achieve H. pylori eradication (two in the BID group and one in the QID group). Therefore, at high eradication rates of 95.9–98.0%, multivariate analysis could not be performed due to statistical error. Instead, we created a new Table 4 using the chi-square test, which is similar to the univariate analysis. In addition, the revised manuscript included the following paragraph: ‘3.4. Analysis of Clinical Factors Affecting H. pylori Eradication’.  

We investigated the various clinical factors that affect H. pylori eradication (Table 4). In relation to H. pylori eradication rate, no significant difference was found in terms of sex, age, BMI, smoking status, cause of the need for eradication, presence of comorbidity, or occurrence of adverse drug events. The H. pylori eradication rate in alcohol drinkers was lower than in non-drinkers; however, this difference was not statistically significant (93.0% vs. 100%, p = 0.076).

  1. Finally, in lines 109-110, text should be corrected, as statistical difference should not be highlighted at non-significant difference p <0.1.

Answer: Thank you for your kind comment. We changed ‘statistical’ and ‘p < 0.1’ to ‘marginal’ difference and ‘p = 0.055’ as below.

Before propensity score matching, a marginal difference was observed between the BID and QID groups regarding current smoking habits (p = 0.055).

Round 2

Reviewer 1 Report

Comments and Suggestions for Authors

Even if retrospective, a calculation of sample size is necessary. If the sample size has not been reached, a justification/explanation is needed.

Author Response

Even if retrospective, a calculation of sample size is necessary. If the sample size has not been reached, a justification/explanation is needed.

Answer: Thank you for your kind comment. Based on your comment, we performed sample size calculation in the first paragraph of ‘2.4. Statistical Analysis’ as below.

The eradication rate of 2-week quadruple therapy was 83% based on our previous study [10]. We assumed a difference of 12% between eradication rates achieved by the two regimens. The minimum sample size was calculated using α- and β-error values of 0.05 and 0.2, respectively. The calculated sample size was 94 in each group, considering a drop-out rate of 10%.

In the Discussion section, we noted that we did not enroll the number of patients based on the sample size calculation.

Thus, the number of patients obtained through sample size calculation was not reached.
